# A Meta-Synthesis of Policy Recommendations Regarding Human Mobility in the Context of Climate Change

**DOI:** 10.3390/ijerph17249342

**Published:** 2020-12-14

**Authors:** Patricia Nayna Schwerdtle, Julia Stockemer, Kathryn J. Bowen, Rainer Sauerborn, Celia McMichael, Ina Danquah

**Affiliations:** 1Heidelberg Institute of Global Health, Universitaetsklinikum Heidelberg, 69120 Heidelberg, Germany; julia.stockemer@uni-heidelberg.de (J.S.); rainer.sauerborn@uni-heidelberg.de (R.S.); ina.danquah@uni-heidelberg.de (I.D.); 2Nursing & Midwifery, Faculty of Medicine, Nursing & Health Science, Monash University, Clayton, VIC 3800, Australia; 3Fenner School of Environment and Society, and Research School of Population Health, Australian National University, Canberra, ACT 2601, Australia; kathrynjbowen@gmail.com; 4Institute for Advanced Sustainability Studies, 14467 Potsdam, Germany; 5Melbourne School of Population and Global Health, University of Melbourne, Parkville, VIC 3010, Australia; 6School of Geography, University of Melbourne, Parkville, VIC 3010, Australia; celia.mcmichael@unimelb.edu.au

**Keywords:** climate change, policy, migration, health, governance

## Abstract

Changing mobility patterns combined with changes in the climate present challenges and opportunities for global health, requiring effective, relevant, and humane policy responses. This study used data from a systematic literature review that examined the intersection between climate change, migration, and health. The study aimed to synthesize policy recommendations in the peer-reviewed literature, regarding this type of environmental migration with respect to health, to strengthen the evidence-base. Systematic searches were conducted in four academic databases (PubMed, Ovid Medline, Global Health and Scopus) and Google Scholar for empirical studies published between 1990–2020 that used any study design to investigate migration and health in the context of climate change. Studies underwent a two-stage protocol-based screening process and eligible studies were appraised for quality using a standardized mixed-methods tool. From the initial 2425 hits, 68 articles were appraised for quality and included in the synthesis. Among the policy recommendations, six themes were discernible: (1) avoid the universal promotion of migration as an adaptive response to climate risk; (2) preserve cultural and social ties of mobile populations; (3) enable the participation of migrants in decision-making in sites of relocation and resettlement; (4) strengthen health systems and reduce barriers for migrant access to health care; (5) support and promote optimization of social determinants of migrant health; (6) integrate health into loss and damage assessments related to climate change, and consider immobile and trapped populations. The results call for transformative policies that support the health and wellbeing of people engaging in or affected by mobility responses, including those whose migration decisions and experiences are influenced by climate change, and to establish and develop inclusive migrant healthcare.

## 1. Introduction

Globally, millions of people move in response to or in anticipation of environmental stress every year, and climate change is becoming more important in their decision to migrate. Extensive research has considered the consequences of climate change separately for: (i) human migration, and (ii) human health.

Firstly, thirty years of climate change and migration research indicates that while climate impacts shape the scale and nature of human migration, climate change does not act in isolation to drive mobility [1]. Migration is multi-causal and climate change interacts with a range of political, socio-economic, and local environmental factors to influence patterns of mobility [2,3,4,5,6,7,8,9,10]. Three predominant ways of framing interactions between environmental change and migration include securitization (migrants are perceived as a threat or strain on resources), protection (migrants are seen to lack agency, requiring defense of their human rights), and as an adaptive response to climate risk (migrants have agency) with climate mobility existing on a spectrum of solutions [11]. While climate change generally acts as a threat multiplier, this does not mean there will be mass cross-border migration. Rather, most climate-related migration is south-to-south and internal and climate change can also reduce mobility, as in the case of trapped or immobile populations [8,12,13].

Secondly, the literature on climate change and health is also extensive and increasingly sophisticated yet lacks coverage of contexts where the capacity to address population health is low and exposure to climate risks is high [14,15]. The climate change and human health literature is also underdeveloped in terms of investigating policy and governance processes, as well as the role of different stakeholders in policy development [16].

It is helpful to consider the broad international policy context relevant to the areas of climate-migration and climate-health. A plethora of international agreements incorporate climate change and migration, and climate change and health. Including, the Sustainable Development Goals (SDGs) incorporating the global health target of universal health coverage, which can be attained by mainstreaming migration and health into all policies [17]. In addition to the SDGs, various agreements have been developed to address climate-related displacement. These include the United Nations Framework Convention on Climate Change (UNFCCC) Conference of Parties (COP) 21 [18], the Nansen Initiative and the Platform on Disaster Displacement [19], and the Sendai Framework for Disaster Risk Reduction [20,21]. Two global compacts, one focusing on migration [22] the other on refugees [23], refer to people who move in the context of climate change and provide opportunities for nation-states to implement good migration and health governance. In terms of the policy context relevant to climate change and health, the Paris Agreement presents a firmer stance on the importance of health compared with The Kyoto Protocol, referring to “the right to health” in the preamble. Shifting from this international scale, there are also regional policy contexts that address climate change and human health; for example the Pacific Islands Action Plan on Climate Change and Health [24] and the WHO’s Special Initiative on Climate Change and Health in Small Island Developing States [25].

As the climate crisis accelerates, it is timely to examine how migration might be more effectively and humanely governed [22]. There is concern that people moving in the context of climate and environmental change may slip through the cracks of existing migration protection frameworks, amplifying risks to their health. Yet it is important to understand and address the health of climate-related migrants as well as the health of people who migrate into or remain in sites with climate-related health risks: appropriate policy frameworks and responses are required. This paper presents policy recommendations emerging from 68 publications reporting the findings of research focused on the climate-migration-health nexus. These publications are reviewed and synthesized to identify broader, principle-based guidance at the nexus of climate, migration, and health.

This review aims to identify, analyze, evaluate, and synthesise the policy recommendations in the literature investigating the nexus between climate change, health, and migration. The specific objectives are to: (a) synthesize policy recommendations from empirical evidence about the climate change-migration-health nexus and present key themes; (b) analyse key methods employed and appraise the quality of the evidence; (c) identify gaps in the evidence base.

## 2. Methods

### 2.1. Protocol and Registration

The protocol for this systematic literature review was developed in consultation with the International Office for Migration (IOM), Migration and Environmental and Climate Change Division, and climate change, migration, and health researchers in our networks. The protocol was registered and published online with the international platform PROSPERO (The International Prospective Register of Systematic Reviews) on 29th August 2018 (Registration no: CRD42018095461). For reporting, we applied the RepOrting Standards for Systematic Evidence Syntheses (RoSES) Pro-forma and flow diagram. This framework integrates diverse methodologies and has been developed for environmental management and conservation research [26]. In contrast, the Preferred Reporting Items for Systematic Reviews and Meta-Analyses (PRISMA) guidelines [27] have been created for systematic reviews and meta-analysis of clinical trials. Therefore, PRISMA was not applicable to the present study. The original search was conducted in December 2019 and was updated in August 2020 to capture current evidence.

### 2.2. Definitions

In this review, we used the World Health Organisation’s (WHO) definition of ‘health’ as a state of complete physical, mental and social well-being and not merely the absence of disease or infirmity [28] and extended this well-known definition to include the health determinants of food and water security.

We defined ‘climate change’ for this review as a change in the state of the climate that can be identified by changes in the mean and/or the variability of its properties, and that persists for an extended period, typically decades or longer [29].

We defined ‘migration’ as an overarching concept covering diverse human mobilities. More specifically, as the movement of persons away from their place of habitual place of residence, either across an international border or within a state regardless of legal status, degree of choice, causes of the movement, or length of stay [30]. Possible climate-related mobility responses comprise forced displacement, migration, and planned relocation [31] as well as immobility. Broadly, forced displacement refers to contexts of forced or involuntary movement of people within or across borders; migration refers to the movement of people within or across borders and has an element of choice; and planned relocation refers to the organised movement of people, typically with government support [19]. Population immobility in contexts of climate risks is increasingly discussed [31], yet lacks definitional clarity: it variously refers to ‘trapped’ populations that are unable to move away from sites of climate risk, and voluntary immobility where people are unwilling to leave their homes despite climate risks [32].

### 2.3. Search Strategy and Eligibility Criteria

This systematic literature view includes studies exploring climate variability and any climate hazard or extreme weather event that could be plausibly linked to climate change. Studies were eligible if health was measured or reported, including as a consideration in migration decision-making, or as an impact at origin, en route, or destination. The study populations include individuals, households, and whole communities who engaged in, or were affected by climate-related mobility: i.e., sending communities, mobile people and populations, and host communities. Importantly, studies explicitly addressing immobility in contexts of climate risk are also included. Taken together, this review includes empirical evidence about health and migration in the context of a changing climate. Given that this nexus is interdisciplinary, the included articles originate from diverse research fields such as public health, demography, policy studies, climatology, human geography, and international relations.

With this background and through consultation with a subject librarian, we selected four academic databases: CABI Direct—Global Health (1973 to present); Ovid Medline (1946 to present); PubMed (1966 to present), and Scopus (1970 to present). In addition, we searched Google Scholar applying the same search terms. The test searches we conducted demonstrated that these databases captured the most relevant articles to answer the review question. These databases were complementary, rather than duplicative and enabled an exhaustive search. Reference lists of selected articles were also searched for relevant articles. Grey literature (likely to cover conceptual papers and editorials) was not included in this systematic literature review, because it focused on peer-reviewed research including empirical data. The search strategy was designed to capture primary studies included in important grey literature reports. Four independent authors used the custom systematic literature review software Covidence (Covidence, Melbourne, Australia.) (PNS, JS, KB, CM) to complete the two-stage screening process.

Three concepts formed the basis of the systematic searches using a Population, Exposure, Outcome (PEO) approach (modified PICO). Specifically, Boolean operators “AND” and “OR” were applied to synonyms for the key search teams Climate Change, Migration, and Health as outlined in Table 1. We included Medical Subject Headings (MeSH) and free text in the searches. The timeframe was chosen because few relevant articles were identified prior to 1990 in numerous test searches and the review aimed to capture contemporary literature. The search for this manuscript was updated in August 2020.

Inclusion and Exclusion criteria are outlined in Table 2. The population studied needed to be engaged in or affected by a mobility response and was not limited to ‘mobile populations’ but included sending and host communities. The mobility response could be any type of population movement at the individual, household, or community level including forced displacement, planned relocation, and, also immobility. The exposure had to be linkable to climate change or climate variability. Other environmental hazards (such as geophysical hazards) were excluded. Studies were not excluded due to poor quality nor study type. At least one health outcome (direct or indirect) needed to be included in the results. Social determinants of health (e.g., education) were not considered a health outcome. Only empirical peer-reviewed studies were included within the timeframe, published in English or German.

### 2.4. Literature Selection

Studies were selected through a two-stage blinded process requiring two independent votes to progress to the next stage. The custom software Covidence was used for study selection and captured all decisions and notes of the reviewers regarding how the studies met or did not meet the criterion. The first stage was title and abstract. The second was full-text screening. Disagreements regarding whether to include a study or not were resolved by a third independent reviewer. Further studies were identified by scanning reference lists of relevant articles. The screening process is documented in Figure 1. The main reasons for exclusion in both phases were that the study was not empirical (i.e.,—editorial) and the research did not include all three concepts but rather focused on just one or two. Interestingly, another reason for exclusion was that the health or migration was not human and related to a plant or animal indicating that migration and health issues in the context of climate change pertain to all living things.

### 2.5. Data Extraction

The extraction codes were the recommendations for policy, practice, and further research included in the selected studies. It became clear during the extraction process that the research recommendations were partly dated and had somewhat been filled. Whilst the recommendations for research were extracted and coded, they are not included in this meta-synthesis. We focussed on the recommendations for policy and practice to give practical insights and to meet the objectives of this review. Developing guidance for a potential research strategy on this topic was seen as beyond the scope of this review.

### 2.6. Quality Assessment

This is the second manuscript from an updated systematic literature review, which presented different findings [33]. Therefore, the quality appraisal was complete for 50 of the included studies identified in the original search (December 2018). The additional 18 studies, identified in the updated search (August 2020), were appraised for quality using the same MMAT tool [34]. MMAT sets five quality appraisal questions for five study types: mixed methods, quantitative descriptive, quantitative non-randomised, quantitative randomised control trials, and qualitative studies. Since MMAT cannot be applied to modeling studies, five studies included in this review were not appraised. This did not make a difference to the results because studies were not excluded based on poor quality. The MMAT tool appraises both study quality itself and the reporting.

### 2.7. Data Analysis and Meta-Synthesis

Recommendations for policy and practice in the selected studies were presented in narrative form (text) and were heterogeneous. Therefore, thematic analysis for meta-synthesis was an appropriate analysis method through which to derive common meaning from the various studies. No selected studies were excluded from the analysis as all 68 studies included some form of recommendation. The extracted recommendations (the data set) were uploaded to N-VIVO (Version 12) and coded. We used Braun and Clarke (2006) to thematically analyse the recommendations and identify themes across the dataset [35]. This involved data familiarisation, code generation, theme searches (multiple), theme review, and defining themes. We used a meta-aggregative approach to the analysis that avoids re-interpreting the recommendations and inductive reasoning [36]. All authors were involved in and approved the thematic analysis that resulted in six themes.

## 3. Results

### 3.1. Included Studies

From 2425 studies identified by the search strategy, 68 eligible studies were included in the analysis (Table 3). The number of relevant studies increased significantly from 2012 with 82% of studies published in the last 8 years (Figure 2). The main reason for exclusion was due to the study not focusing on the nexus between climate change, migration, and health, but rather limited to climate change and migration (without health) or climate change and health (without migration). Studies were also excluded because they did not present empirical evidence, or investigated health and migration of animal or plant species. The original systematic literature review search was first conducted in December 2018 and a publication focused on another data-set was published in Environmental Review Letters [33]. The search was updated in August 2020, capturing 18 additional studies using the same protocol. This publication focuses on the recommendation for policy, practice, and further research extracted from the included studies.

### 3.2. Study Settings

Most selected studies were conducted in Bangladesh (*n*: 19, 24%), the USA (*n*: 8, 12%) and India (*n*: 5, 7%). Studies covered all WHO regions, including South East Asia (36%), Africa (26%), Pan America (22%), Western Pacific (9%), Eastern Mediterranean (3%), and Europe (4%). There were 81 study settings overall from 68 selected studies as depicted in Figure 3.

### 3.3. Quality Appraisal

Overall, the quality of the included studies was high with the quantitative non-randomised studies (*n*: 12) achieving the highest quality assessment result, with 67% scoring between ‘good’ and ‘very high’ and no studies being appraised as poor (Figure 4). The quality of quantitative descriptive studies (*n*: 17) was also high overall, with 71% of studies scoring between ‘good’ and ‘very high’. Overall, the qualitative studies (*n*: 17) were appraised as high quality with 65% scoring between ‘good’ and ‘very high’. The quality of mixed methods studies (*n*: 17) was also high overall with 65% attaining a ‘good’ or ‘high’ quality rating, yet no mixed methods studies achieved a ‘very high’ quality score according to MMAT.

### 3.4. The Links between Climate Mobility and Health

A range of climate hazards (*n*: 24) were associated with diverse mobility responses and health outcomes studied, without clear trends or patterns emerging. The climate hazards in the studies were half sudden-onset (*n*: 12) such as storms, floods, and cyclones, and half slow-onset (*n*: 12) such as drought, sea-level rise, and glacial retreat. The most studied hazards included floods (*n*: 19; 16%), rainfall variability (*n*: 18; 15%), drought (*n*: 17; 14%) and multiple climate hazards or general climate change (*n*: 14; 12%). Other common climate hazards included extreme heat, sea-level rise, and hurricanes (*n*: 6; 5%). The predominant mobility responses were forced displacement, relocation (planned and forced), seasonal migration, and rural-urban migration. The predominant health outcomes studied were food and water security, access to healthcare services, mental health issues, and infectious disease [33].

### 3.5. Thematic Analysis of Policy Recommendations

Six themes were identified in the thematic analysis of policy recommendations extracted from 68 studies. A synopsis of the themes with illustrative quotes are outlined in Table 4.

## 4. Discussion

This review deepens our understanding around the complex mechanisms through which climate change impacts contribute to migration and health outcomes. This review reveals some useful guidance for migration and health policy and practice in the context of climate change. The geographical settings where this nexus research has taken place to date, the predominant study designs, their quality, and the relationships between climate hazards, mobility responses, and health outcomes are discussed below. Finally, a narrative synthesis of the policy recommendations extracted from the included studies is placed in the context of the current policy environment on this topic.

### 4.1. Study Settings

The predominant geographical foci of the studies in this review; Sub-Saharan Africa, South Asia, and Latin America. Bangladesh is of particular interest to climate change, migration, and health studies possibly, because it is highly exposed and vulnerable to climate hazards, densely populated.

These regions identified in the research align with existing evidence that indicates that currently, most climate-related migration is internal and takes place in developing countries [31,102]. Three regions in the world are projected to see more than 140 million internal climate-related migrants by 2050, without urgent global and national climate action and economic development [102].

### 4.2. Quality Appraisal and Study Design

Surveys and case reports were a common design of the included quantitative descriptive studies (*n*: 17). Some of these studies looked at the association between climate hazards, migratory behaviours, and health outcomes and sought to determine the extent to which migration was adaptive or maladaptive. Overall, the quality of quantitative descriptive studies was high with 71% scoring between ‘good’ and ‘very high’ (see Figure 4). Analysis ranged from simple descriptive statistics to regression analyses.

Cross-sectional analytical studies were a common quantitative non-randomised study design that compared groups to examine the impact of climate hazards on health (*n*: 12). These types of studies compared mobile and non-mobile households or communities. Others compared households at variable levels of climate risk concerning health. Overall, the quality of these quantitative non-randomized studies was high with 67% of studies scoring between ‘good’ and ‘very high’. This was the only category of studies without any scoring ‘poor’ quality. Quality issues included mismatching of temporal-spatial scales of the hazard, mobility response, and health outcomes and inadequate demographic matching of comparison groups.

Case studies and narrative research were prevalent in the selected studies (*n*: 17). These studies used qualitative methods to unpack health concerns in settings where climate hazards were affecting people’s mobility patterns. Overall, the quality was high with 65% of studies scoring between ‘good’ and ‘very high’, although these studies often incorporated small sample groups with limited ability to generalise.

Mixed methods (*n*: 17) studies included in this review commonly combined surveys and interviews or focus groups to explore the experiences of mobile/displaced people facing climate risks, and the consequences for health. The quality was high overall with 65% attaining a ‘good’ or ‘high’ rating with no studies achieving excellence according to Mixed Methods Assessment Tool (MMAT) criteria. Poorer scores were related to either the quantitative or qualitative component scoring higher and less transparent integration of results and interpretation of both methods. Overall, a more meaningful and transparent integration of climate data would have more clearly linked the mobility responses and health outcomes to the climate scenario [33].

### 4.3. Climate Mobilities and Health

The relationships between climate change, mobility, and health are complex and connections are population-specific and vary over space and time [33]. This review covers these relationships published in the peer-reviewed literature, which focuses on infectious diseases, access to healthcare, mental health, and food insecurity. While the global burden of disease has shifted to non-communicable diseases (NCD’s), NCDs are an under-researched theme with a greater focus on infectious disease in climate-migrant populations. The review also reinforces the recognition that migration does not capture the diverse ways in which people do and do not move in response to a changing climate. Highlighting the need for researchers to challenge the ‘climate change causes mass human migration’ narrative and to shift attention from climate migration to more diverse forms of climate mobility and indeed immobility [103].

### 4.4. Thematic Analysis of Policy Recommendations

The effect of climate-related mobility on health depends on the policy decisions made by host, home, and transit states and involved organisations, rather than on the mobility itself [104] highlighting the value of evidence-based policy in migration and health governance. In light of this, we extracted and analysed the policy recommendations from all 68 included studies. Six themes were identified that are presented graphically with predominant codes in Figure 5. The themes range from overarching recommendations such as avoiding universal promotion of migration or supporting community participation in migrant health initiatives, to targeted recommendations such as specific climate-change adaptation activities or the importance of preserving social and cultural ties in contexts of climate mobility.

Firstly, there were consistent recommendations to avoid universal promotion of migration as an adaptive response to climate risk, to prevent forced migration by investing in climate change adaptation at origin, and to consider relocation only as a last resort [12,43,55,58,70,76,85,97]. These recommendations echo the broader calls by, for example, the International Organization of Migration (IOM) to minimize forced climate-related migration [11] and also the need to avoid assumptions that mobility is inherently positive or negative [73]. Given widespread preferences to remain in sites of belonging, many studies reviewed here called for policy initiatives that enable people to cope with, avoid and prevent the impacts of climate change at origin to prevent forced migration [105].

The included studies made recommendations to support investment in climate change adaptation and supporting the sustainable development of agriculture generally. Specific suggestions included providing credit facilities and building agricultural extension services—also known as agricultural advisory services—that build knowledge of agronomic techniques and skills to improve productivity, food security, and livelihoods. The focus was clearly on rural farming communities and adaptation with no studies exploring mitigation and few urban settings. This raises some questions about the extent to which the research is focusing on adaptation in a critical period when the mitigation window is closing. Alternatively, this gap may simply reflect that when migration is triggered, there is an urgent adaptation situation in play. In some contexts, existing migration flows and networks may provide an opportunity for investment in the economies of sending communities (via remittances) that can increase opportunities for in situ adaptation and resilience among those who remain.

The second theme highlighted the importance of the preservation of cultural and social ties for the health of mobile populations through preserving and revitalizing traditional solidarity measures [55,62,74,75,76,94]. Selected articles recommended strengthening governance or socio-ecological systems, favouring community-led approaches, and in doing so, recognizing the agency and inherent resilience of communities and to promote collaborative, adaptive migration governance structures. This agency and resilience focus is predominant in research recommendations yet somewhat lacking in research questions whereby disease and risk factors are the focus, rather than positive health outcomes and protective factors.

The third theme pertained to policy recommendations that sought to enable migrants to participate socially and economically in destination sites. The focus was on employment and income that have clear advantages for the health and well-being of both migrant and host populations as well as sending communities if remittances are mobilized [38,54,55,56,57,59,65,67,88,89,91,96]. These studies tended to focus on subsistence farmers in rural settings, although many other types of populations will have migration decisions influenced by climate change, and the causal pathways can appear over-simplified. This theme incorporated studies about relocation and resettlement and was the theme with the most focus on host and sending communities rather than purely mobile communities. A well-defined example of promoting self-sufficiency was planned relocation in the Carteret Islands, where there was a recognition that the resettled Carteret families may not have the skills necessary to cultivate kitchen gardens. New arrivals received training inappropriate agricultural techniques, enabling self-sufficiency through income generated from selling cash crops [55].

The fourth theme brought out the recommendations around the need to strengthen health systems generally where migrants are (in both sending communities and destination areas) in terms of both primary health care and more specialized vertical programs such as for HIV and Maternal Child Health (MCH) services. The findings in the selected articles revealed financial, geographic, and cultural barriers for migrants accessing healthcare in the context of climate change and led to recommendations to reduce or remove these barriers to improve migrant health for example by including migrants in health insurance schemes [32,42,46,59,60,61,62,64,70,73,74,79,88,89,90,92,100]. There are clear benefits to broader population health from investing in health systems strengthening approach, so the recommendations within this theme would have substantial flow-on benefits to the community at large.

The fifth theme went upstream from quality accessible health services, to identify the importance of ensuring basic requirements for health such as food, water, and shelter, which are basic human rights and necessary to protect life, reduce suffering and preserve human dignity. These studies highlighted the need to focus on health equity in a range of settings including climate-vulnerable regions and sites of relocation and resettlement. This includes the need to integrate migrants into labour markets (see Theme 3) to support livelihoods and food security and to enable access to education [44,45,53,57,64,67,69,72,90,91,94].

This theme corroborates the demand for establishing migrant-inclusive health systems, as suggested by the Lancet Commission on Migration and Health [106]. They constitute the basis for the supply and utilization of patient-centered access to health and social protection. This theme reiterates WHO’s Global Action Plan for promoting the health of refugees and migrants under Priority 4. Enhance capacity to tackle the social determinants of health and to accelerate progress towards achieving the Sustainable Development Goals, including universal health coverage.

Finally, the sixth theme focused on the need for policy to integrate health into the full range of loss and damage calculations [32,55,60,61,63,100]. Loss and damage refer to the negative effects of climate variability and change that people are not able to cope with or adapt to. Research and policy discussions of loss and damage recognize that climate change impacts are differential, with greater losses accruing to vulnerable populations and regions, thereby exacerbating inequities [107]. And it is important to note that vulnerability to climate change is fundamentally a matter of political economy, with those least responsible for climate change most at risk from adverse climate impacts. For example, immobile populations in sites of climate risk may be trapped and experience loss and damage including because they lack resources, assets, and networks to enable migration away from sites of risk [32].

There is critical need and value in initiatives that address ‘vulnerability’ and improve ‘adaptive capacity’ by investing in adaptation and human development in local sites of climate risk, thereby potentially limiting the need for out-migration. However, policy initiatives that focus on ‘the vulnerable’ and proximate social and environmental contexts are at risk of obscuring complex power relations and global inequities that create these vulnerabilities or limit adaptive capacity [108].

Policy recommendations referred to both economic and non-economic losses and damages and suggested widening our understanding of the linkages between mobility responses (including immobility) and wellbeing by looking at non-economic loss and damage and its links to mental health. It was argued that a lack of focus on this aspect might constitute a potentially costly public health inaction. The need identified was to provide culturally appropriate compensation for displaced and host populations with a range of populations at heightened risk requiring careful consideration including women and girls, elderly people, trapped populations, people living with disabilities, and people living with HIV/AIDS.

Immobile populations living in such contexts may experience adverse health impacts that emerge from changes in water and food security, disease ecology, flooding and saltwater intrusion, and the psychosocial impacts of disrupted livelihoods [109]. These represent important aspects of loss and damage that emerge from climate change impacts [110]. Despite a widespread focus on livelihood security and damage to physical assets as key aspects of loss and damage, adverse health impacts among climate-affected populations—such as food and water insecurity—also represent aspects of loss and damage. These health costs and consequences, including among (im)mobile populations in contexts of climate risk, cause significant harm, and impede sustainable development.

The studies we review here all discuss the policy and practice significance of their findings. Most note that the scale of climate-related migration is projected to increase in the coming decades (while also noting that climatic factors rarely act alone in shaping human mobility) and that there are health risks and opportunities of climate-related mobility. Climate change is likely to act as an amplifier of health risks among mobile populations, rather than creating new vulnerabilities that require distinct policy and practice responses. Nevertheless, it is still an important and increasingly significant consideration in both migration and health policy. Therefore, it will remain of importance and value to strengthen the migrant sensitivity of health systems, ensure universal health coverage, and continue efforts to address the social determinants of health inequities for all including for mobile populations. Yet there may also be a need to respond to the increasing vulnerability of some migrants. There will likely be climate-related mobility across and within borders of countries in the Global South where health systems are generally weaker and have a lower capacity to deal with increased demand. And many people migrating in the context of climate change may move into areas of increased risk, such as in Bangladesh where informal settlements form in flood zones [65,79]. So, rather than creating new categories of vulnerable mobile populations, there will be an amplified need to address the health of mobile populations in often under-resourced and climate-vulnerable sites.

### 4.5. Limitations

In terms of study design, we used a priori protocol and a double-blinded approach for study selection and quality appraisal to limit bias. In addition to language and publication bias, there is always a risk in any systematic literature review—despite a thorough search strategy—that articles are missed. Studies were not omitted due to quality as per the MMAT tool recommendations [34], which may have led to findings of studies with weaker designs being included in the meta-synthesis. Five studies did not fit within the MMAT categorisation (modelling studies) and were therefore not assessed for quality, which remains a limitation of this review. While grey literature was not included, the focus on peer-reviewed empirical evidence was considered a strength.

There are limitations when considering the implications of this study. Considering health research is context and population-specific and climate-related migration is variable over space and time. Therefore, translating or generalising these findings to a range of diverse settings and populations may present risks. There is a need to understand the risks, exposures, vulnerabilities, and capacities of unique populations and settings to inform policy and practice decisions.

Finally, some included studies suggested (rather than demonstrated) potential (rather than actual) links to climate change. This may have led to an overestimation of the relationship between climate change, and health and mobility responses.

### 4.6. Future Research Directions

The recommendations extracted, analysed, and synthesized in this review pertained mainly to rural livelihoods revealing a potential gap in researching this nexus in urban settings. The research focused on mobile populations and less so on the entire migration ecosystem including sending and host communities echoing a call for research on climate mobilities to shift part of its focus from climate-sensitive sending areas to destination areas [103]. Further, there was a focus on adaptation with less attention paid to mitigation activities. It may be worthwhile looking at these concepts together, or on a spectrum whereby adaptation activities can play a role in mitigation and vice versa. There was very little to no reference to conflict in these studies although we know that conflict and migration are both related socially mediated, tertiary impacts of climate change and likely to be playing a role in some of these settings and influencing health and health resources. Finally, the included studies focused on agricultural climate change adaptation with less attention paid to infrastructure and health.

## 5. Conclusions

As climate change continues to shape patterns and scales of human migration, policymakers are challenged to make evidence-based decisions that enable competent governance of orderly, safe, regular, and humane migration, that safeguards human health and wellbeing. This review identified, analysed, and synthesized what is known to date from research investigating the climate change, migration, and health nexus. The findings largely confirm principles of good migration governance; that the universal promotion of migration should be avoided because it is not always adaptive, that forced migration should be prevented and that mobile populations should be supported in their decision-making. Some elements of our current understanding are reinforced in terms of the need to favour community-led approaches, provide durable solutions, preserve cultural ties and to enable migrant participation because it has the potential to maximise benefit for all those affected by mobility responses. This review also draws attention to more novel discussions such as non-economic loss and damages arising from climate change that include health impacts. The included studies focused on adaptation without mention of the role of mitigation, or adaptive responses that also mitigate climate change. Revealing an area where the climate, migration, and health community may want to take a stand, in the spirit of true prevention. The research also focused on rural rather than urban settings and agriculture and less so health. Yet the policy recommendations synthesized by this review still call for the provision of basic prerequisites for health, a focus on health equity, and access to health care, reiterating that these fundamental requirements for health are not a reality for all. In sum, the results call for transformative policies that support the health and wellbeing of people engaging in and affected by mobility responses, including those whose migration decisions and experiences are influenced by climate change, and to establish and develop inclusive migrant healthcare.

## Figures and Tables

**Figure 1 ijerph-17-09342-f001:**
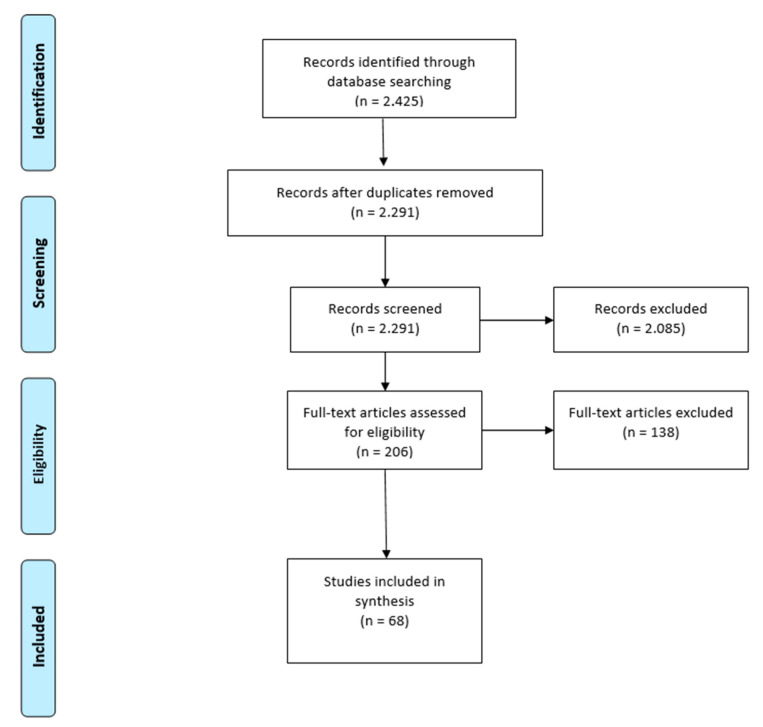
Screening chart.

**Figure 2 ijerph-17-09342-f002:**
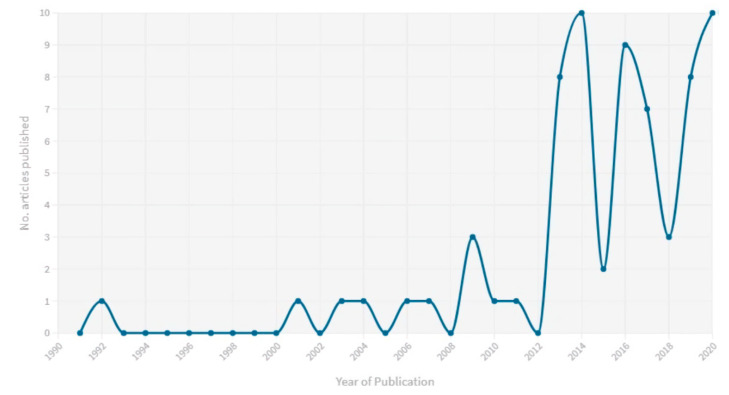
Publication year of included studies.

**Figure 3 ijerph-17-09342-f003:**
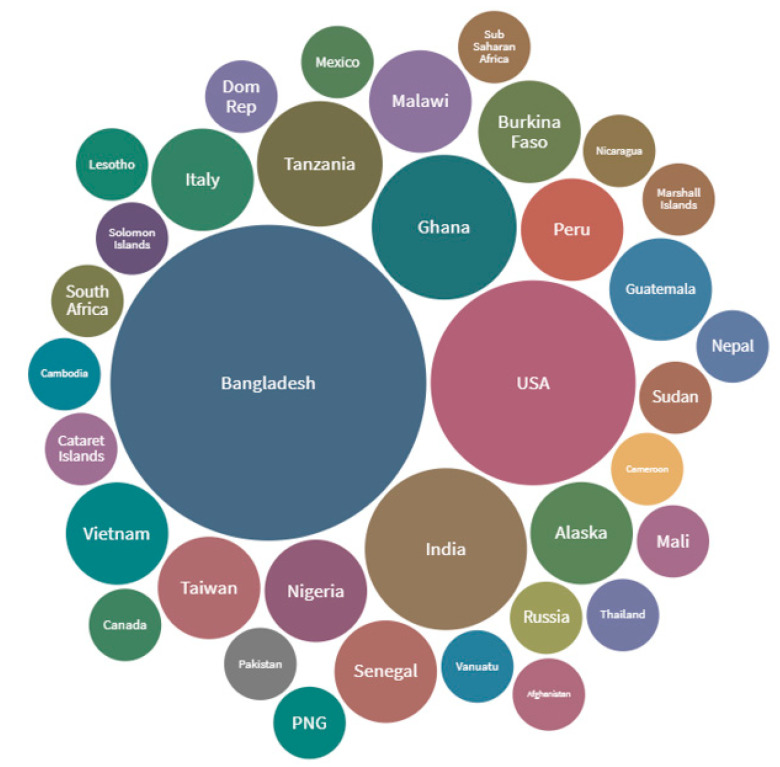
Study Settings. The size of bubbles indicates how many of the 68 studies were conducted in each study site (details in text).

**Figure 4 ijerph-17-09342-f004:**
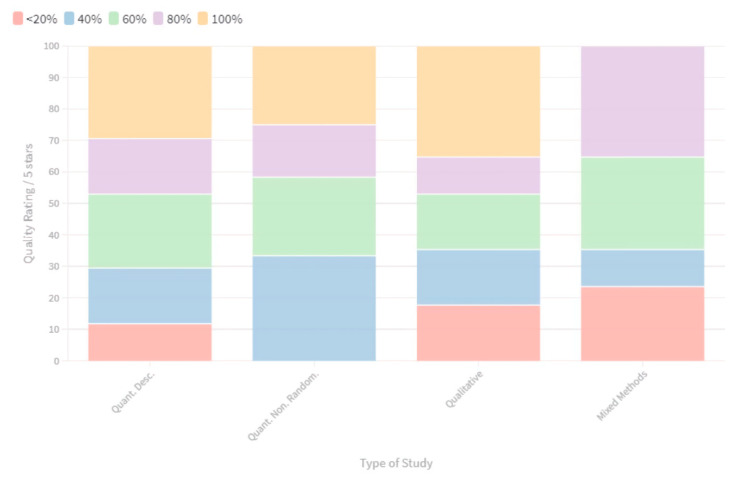
Quality Appraisal of included studies with MMAT: The majority of studies (approx. 70%) were of moderate—high quality according to the quality appraisal tool. Individual Quality Appraisal. The ranking system is according to the articles adherence to each of the MMAT 5 question criteria i.e.,—frequency (%) we could answer ‘yes’ to the quality appraisal question and not ‘no’ or ‘cannot tell’. (100–81% Very high, 80–61% High, 60–41% Good, 40–21% Fair, <20% Poor).

**Figure 5 ijerph-17-09342-f005:**
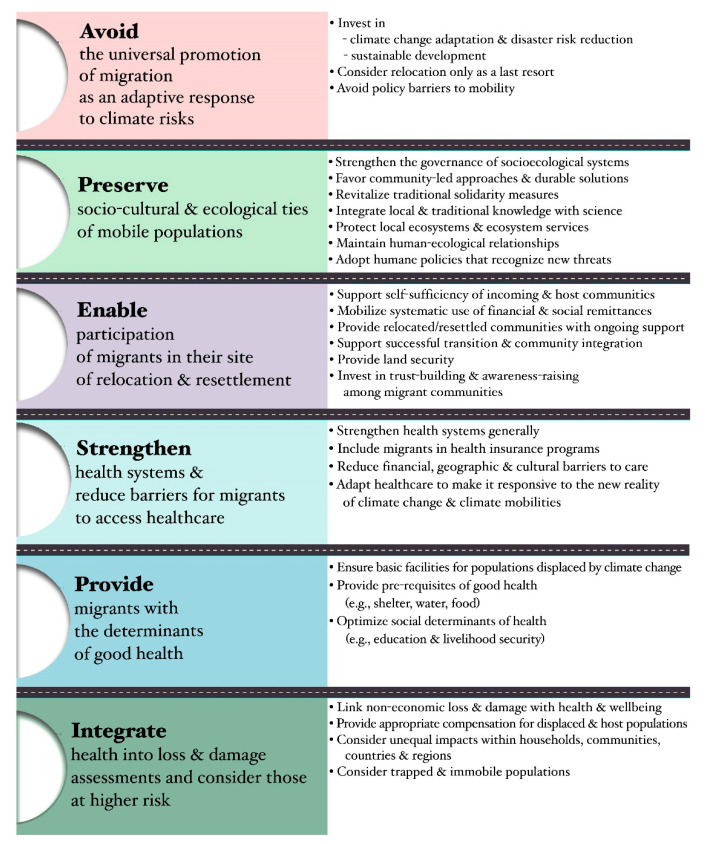
Policy recommendations synthesis.

**Table 1 ijerph-17-09342-t001:** Search Terms.

Migration (Population)	Climate Change (Exposure)	Health (Outcome)
Population movement Displacement Population displacement Forced displacement Internal displacement Seasonal migration Permanent migration Planned relocation Migrant Mobility Internally Displaced Persons Refugee	Climate variability Global warming Weather variability Greenhouse effect Sea level rise Environmental disaster Natural disaster Drought Climate hazard	Well being Disease: NCD, Communicable, Infectious Epidemiolog Lifestyle Co-benefits Mortality Morbidity Climate sensitive disease Nutrition: Malnutrition, Undernutrition Psychosocial Dehydration Health Services: Water and Sanitation, Food Security

**Table 2 ijerph-17-09342-t002:** Inclusion and Exclusion Criteria.

Criterion	Inclusion	Exclusion
Population	Engaging in/affected by mobility response. Any country. Any population.	Not human mobility: plant or animal.
Exposure	Climate change, variability, or natural hazard influencing mobility responses.	Other migration drivers (e.g., political, economic, social, demographic) with no reference to climate or environmental change. Hazard not relatable to climate change (e.g., volcano/earthquake)
Outcome	Health-related: either a direct measure of health outcome (i.e.,—disease prevalence) or an indirect measure of health (i.e., food security and water/sanitation/hygiene—[WaSH]). Access to healthcare. Psychosocial health.	No health outcome. Not human health; animal/plant. Social determinants of health; income, livelihood, employment, education.
Nexus	Includes climate change, migration, and health.	Focus on two elements (dyad) of the nexus (i.e., climate-health, climate-migration). Focus on OneHealth, Planetary Health, Environmental Health.
Study Type	Peer-reviewed empirical research. All designs: quantitative, qualitative, mixed methods, modeling.	Not empirical, systematic review, viewpoint, editorial, book chapter, grey literature, dissertation, conference proceeding, report.
Time	1990–2020 (August).	Outside this timeframe.
Language	Full text available in English or German.	Other languages.

**Table 3 ijerph-17-09342-t003:** Overview of selected studies with quality appraisal.

Reference	Title	Research Setting	Focus of the Study	Study Design and QA
Abah and Petja (2016) [37]	Assessment of potential impacts of climate change on agricultural development in the Lower Benue River Basin	Nigeria & Cameroon	(Mobility) Rural-Urban migration. (Climate) Rainfall & temperature variability, floods & droughts, heat stress, surface water trends. (Health) Infectious disease, HIV/AIDS.	Modelling Unable to assess using MMAT
Adams (2016) [12]	Why populations persist: mobility, place attachment and climate change	Peru	(Mobility) Trapped populations/immobility. (Climate) Temperature extremes, excessive precipitation, abrupt seasonal weather changes & drought, glacial retreat. (Health) Temperature extremes, excessive precipitation, abrupt seasonal weather changes & drought, glacial retreat.	Quantitative. Analytical: Cross-sectional survey ***
Afifi, Liwenga and Kwezi (2014) [38]	Rainfall-induced crop failure, food insecurity and out-migration in Same-Kilimanjaro, Tanzania	Tanzania	(Mobility) Seasonal migration & temporal migration. (Climate) Rainfall variability, floods & droughts, water shortages. (Health) Food insecurity.	Mixed methods; Expert interviews & quantitative descriptive survey. Participatory research approach **
Ahmed, Kelman… Shamsudduha (2019) [39]	Indigenous people’s responses to drought in northwest Bangladesh	Bangladesh	(Mobility) Seasonal migration. (Climate) Drought. (Health) Food security, Water security, Water-borne disease.	Mixed methods: Household survey & participatory rural appraisal (qualitative). ****
Albert, Bronen… Grinham (2018) [40]	Heading for the hills: climate-driven community relocations in the Solomon Islands and Alaska provide insight for a 1.5 °C future	Solomon Islands & Alaska	(Mobility) Relocation: Supported and Unsupported. (Climate) Sea level rise, reduced Arctic sea ice, melting permafrost, sea-level rise, erosion & flooding. (Health) Access to health care facilities, health & safety risks, water borne disease, vector borne disease, dietary adaptation.	Qualitative case studies ****
Amstislavski, Zubov… Weedon (2013) [41]	Effects of increase in temperature and open water on transmigration and access to health care by the Nenets reindeer herders in northern Russia	Northern Russia	(Mobility) Inhibition (delaying) regular transmigration. (Climate) Temperature increase, reduction of ice-rich permafrost & glaciers, changes in hydrological cycles. (Health) Access to health care facilities, risks of injury.	Quant. non-rand. Water body & temp. data, migrant health care records. ***
Anastario, Shehab, and Lawry (2009) [42]	Increased GBV Among Women Internally Displaced in Mississippi 2 Years Post–Hurricane Katrina	Mississippi, USA	(Mobility) Forced displacement. (Climate) Hurricane. (Health) Sexual & physical violence, suicidal ideation & attempts, depression.	Quant. non-rand. Successive cross-sectional surveys. *****
Anupama, Deb…Vajjha (2016) [43]	Seasonal Migration and Moving Out of Poverty in Rural India: Insights from Statistical Analysis	Rural India	(Mobility) Temporary/seasonal migration. (Climate) Drought. (Health) HIV/AIDS, water-borne disease, general & sexual health, social issues.	Quant. non-rand. **
Assan, Caminade and Obeng (2009) [44]	Environmental variability & vulnerable livelihoods: Minimising risks & optimising opportunities for poverty alleviation	North Eastern Ghana	(Mobility) Temporary migration, circular migration. (Climate) Erratic and declining mean rainfall. (Health) Food insecurity.	Mixed methods. 1° & 2° quantitative/qualitative. ***
Atta Ur Rahman, Akhtar and Siddiqui (2013) [45]	Psychological Effects among Internally Displaced Persons (IDPS) residing in two districts of Sindh	Sindh, Pakistan	(Mobility) Forced displacement. (Climate) Flood. (Health) Mental disorders.	Descriptive cross-sectional study. *
Ayeb-Karlsson, Kniveton, Cannon (2020) [32]	Trapped in the prison of the mind: Notions of climate-induced (im)mobility decision-making and wellbeing from an urban informal settlement in Bangladesh	Bangladesh	(Mobility) Immobility. (Climate) Multiple climate-hazards. (Food) Sense of belonging and mental health.	Q-methodology & discourse analysis. ****
Baker (2020) [46]	Climate change drives increase in modeled HIV prevalence	Sub Saharan Africa	(Mobility) Migration. (Climate) General climate change and temperature. (Health) HIV.	Modelling
Bayar and Aral (2019) [47]	An Analysis of Large-Scale Forced Migration in Africa	African region	(Mobility) Forced Migration. (Climate) Multiple climate-hazards. (Health) Public health and wellbeing as a component of human security.	Modelling
Behr and Diaz (2013) [48]	Disparate Health Implications Stemming from the Propensity of Elderly & Medically Fragile Populations to Shelter in Place During Severe Storm Events	North Carolina, USA	(Mobility) Forced displacement, immobility. (Climate) Hurricane. (Health) Access to the support system, medical records, medical regimens, nutrition.	Quant. desc. **
Carney and Krause (2020) [49]	Immigration/migration & healthy publics: the threat of food insecurity	USA, Dominican Republic, Italy	(Mobility) International migration. (Climate) Climate change. (Health) Food Insecurity.	Qualitative: Ethnography. **
Chen, Lai… Chen (2011) [50]	Risk factors for PTSD after Typhoon Morakot among elderly people in Taiwanese aboriginal communities	Taiwan	(Mobility) Relocation. (Climate) Typhoon. (Health) PTSD, injury/death, self-perceived health.	Quant. desc. *****
Coker, Hanks… Franzini (2006) [51]	Social and Mental Health Needs Assessment of Katrina Evacuees	Houston, USA	(Mobility) Forced displacement/evacuation. (Climate) Hurricane. (Health) NCDs, PTSD.	Quant. desc. **
Craven (2015) [52]	Migration-affected change and vulnerability in rural Vanuatu: Migration-affected change in rural Vanuatu	Vanuatu	(Mobility) Seasonal migration. (Climate) Rainfall variability. (Health) Health financing, food security.	Qualitative; Interviews & focus groups. ***
Di Giorgi, Michielin and Michielin (2020) [53]	Perception of climate change, Loss of social capital and mental health in two groups of migrants from African countries	Italy	(Mobility) International migration. General climate change. (Health) Social Capital. Mental Health.	Quant. non-rand. Semi-structured interviews. **
Dinkelman (2017) [54]	Long-Run Health Repercussions of Drought Shocks: Evidence from South African Homelands	South Africa	(Mobility) Multiple types of migration, internal migration, labour migration. (Climate) Drought. (Health) Disability (visual, hearing, speech, mental, physical).	Quant. non-rand. Analytical. ***
Edwards (2013) [55]	The Logistics of Climate-Induced Resettlement: Lessons from the Carteret Islands, Papua New Guinea	Carteret Islands, PNG	(Mobility) Forced displacement, relocation. (Climate) Sea-level rise, King tides, storm surge, floods. (Health) Food insecurity, mental health.	Qualitative; Case study interviews. *
Etzold, Ahmed…Neelormi (2014) [56]	Clouds gather in the sky, but no rain falls. Vulnerability to rainfall variability and food insecurity in Northern Bangladesh and its effects on migration	Northern Bangladesh	(Mobility) Labour migration (permanent, seasonal, temporary), immobility. (Climate) Rainfall variability. (Health) Food insecurity.	Mixed Methods. Interviews. Focus Groups. Questionnaires. *
Grawert (1992) [57]	Impacts of male outmigration on women: A case study of Kutum/Northern Darfur/Sudan	Western Sudan	(Mobility) Out-migration. (Climate) Drought. (Health) Food insecurity.	Qualitative; Case study. *
Grecequet, DeWaard…Abel (2017) [21]	Climate Vulnerability and Human Migration in Global Perspective	Global	(Mobility) Multiple types of migration. (Climate) Multiple climate-hazards. (Health) Mortality from climate-sensitive diseases, vector-borne disease, health (together with food, water, ecosystem services).	Modelling
Gautam (2017) [58]	Seasonal Migration and Livelihood Resilience in the Face of Climate Change in Nepal	Nepal	(Mobility) Seasonal migration, labour migration. (Climate) Drought, rainfall variability. (Health) Food insecurity.	Mixed Methods ****
Haque, Parr, Muhidin (2019) [59]	Parents’ healthcare-seeking behavior for their children among the climate-related displaced population of rural Bangladesh	Bangladesh	(Mobility) Forced Displacement. (Climate) Multiple climate-hazards. (Health) Child health care. Parental health-seeking behaviour.	Quant. non-rand. Analytical. *****
Haque, Parr, Muhidin (2020) [60]	Climate-related displacement, impoverishment and healthcare accessibility in mainland Bangladesh	Bangladesh	(Mobility) Forced Displacement. (Climate) Flood and riverbank erosion. (Health) Access to health care. Access to WASH.	Quant. non-rand. Analytical. *****
Haque, Parr, Muhidin (2020) [61]	The effects of household’s climate-related displacement on delivery and postnatal care service utilization in rural Bangladesh	Bangladesh	(Mobility) Forced Displacement. (Climate) Multiple climate-hazards. (Health) Delivery at a health centre. Post-natal care service utilization.	Quant. non-rand. Analytical. *****
Heaney and Winter (2016) [62]	Climate-driven migration: an exploratory case study of Maasai health perceptions and help-seeking behaviour	Tanzania	(Mobility) Rural - urban migration. (Climate) Drought. (Health) Help seeking behaviour, health care utilisation, food insecurity, water insecurity.	Qualitative; Case study with interviews. *****
Hori and Schafer (2010) [63]	Social costs of displacement in Louisiana after Hurricanes Katrina and Rita	Louisiana, USA	(Mobility) Forced Displacement/evacuation. (Climate) Hurricane. (Health) Access to primary health care.	Quant. non-rand. Cross-sectional retrospective. ***
Hunter and Simon (2017) [64]	Might climate change the “healthy migrant” effect?	Mexico & USA	(Mobility) International migration. (Climate) Rainfall variability. (Health) Self-assessed health, adult height (early life nutritional & health conditions).	Quant. non-rand. Retrospective. **
Hutton and Haque (2003) [65]	Patterns of Coping and Adaptation Among Erosion-Induced Displacees in Bangladesh: Implications for Hazard Analysis & Mitigation	Bangladesh	(Mobility) Forced displacement. (Climate) Riverbank erosion. (Health) Psychological distress.	Quant. desc. *
Hutton and Haque (2004) [66]	Human Vulnerability, Dislocation and Resettlement: Adaptation Processes of River-bank Erosion-induced Displacees in Bangladesh	Bangladesh	(Mobility) Involuntary migration, erosion-induced displacement, rural-urban migration. (Climate) Flooding/river-bank erosion. (Health) Health problems, household hunger.	Quant. desc. ***
Iqbal, Donjadee… Liu (2018) [67]	Farmers perceptions of and adaptations to drought in Herat Province, Afghanistan	Herat province, Afghanistan	(Mobility) Labour migration. (Climate) Drought. (Health) Food insecurity, malnutrition.	Quant. desc. ****
Islam and Hasan (2016) [68]	Climate-induced human displacement: a case study of Cyclone Aila in the south-west coastal region of Bangladesh	Bangladesh	(Mobility) Forced displacement. (Climate) Cyclone. (Health) Food insecurity, malnutrition.	Mixed Methods *
Islam, Sallu… Paavola (2014) [69]	Migrating to tackle climate variability and change? Insights from coastal fishing communities in Bangladesh	Bangladesh	(Mobility) Rural – Urban migration (Island to mainland). (Climate) Climate variability. (Health) Physical fitness, access to WASH.	Mixed Methods ***
Jacobson (2019) [70]	When is migration a maladaptive response to climate change?	Cambodia	(Mobility) Migration. (Climate) Rainfall variability, drying, increased mean average temperature. (Health) Food security.	Quant. desc. *****
Kabir, Rahman… Milton (2016) [71]	Climate change and health in Bangladesh: a baseline cross-sectional survey	Bangladesh	(Mobility) Forced displacement. (Climate) Cyclones, floods, salinity. (Health) Infectious diseases, malaria, dengue, pneumonia, diarrhoea, height & weight, access to health care facilities.	Quant. non-rand. **
Loebach and Korinek (2019) [72]	Disaster vulnerability, displacement, and infectious disease: Nicaragua and Hurricane Mitch	Nicaragua	(Mobility) Forced displacement. (Climate) Hurricane. (Health) Infectious disease.	Quant. non-rand. *****
Loevinsohn (2015) [73]	The 2001-03 Famine and the Dynamics of HIV in Malawi: A Natural Experiment	Malawi	(Mobility) Rural—urban migration. (Climate) Change change and variability. (Health) HIV.	Quant. desc. Retrospective natural experiment ****
Low, Frederix… Schwitters (2019) [74]	Association between severe drought & HIV prevention & care behaviors in Lesotho: A population-based survey	Lesotho	(Mobility) Internal, labour and circular migration. (Climate) Drought. (Health) HIV.	Quant. desc. *****
McElfish, Moore… Peter (2016) [75]	Social Ecology and Diabetes Self-Management among Pacific Islanders in Arkansas	Arkansas, USA	(Mobility) Migration. Climate Change. (Health) Type 2 diabetes (language, translation, treatment, diabetes self-management).	Qualitative; Case study *****
Mertz, Mbow… Diouf (2009) [76]	Farmers’ Perceptions of Climate Change and Agricultural Adaptation Strategies in Rural Sahel	Senegal (savanna)	(Mobility) Migration. (Climate) Climate variability (wind, rain, dust storms). (Health) Health, reduced solidarity.	Qualitative; Focus group *****
Messias and Lacy (2007) [77]	Katrina-Related Health Concerns of Latino Survivors and Evacuees	USA	(Mobility) Evacuation. (Climate) Hurricane. (Health) Hunger, environmental health risks, sleep disturbance, access to healthcare.	Qualitative; Narrative research *****
Milan and Ruano (2014) [78]	Rainfall variability, food insecurity and migration in Cabricán, Guatemala	Guatemala	(Mobility) Seasonal & permanent migration. (Climate) Rainfall variability. (Health) Food security (threat of local livelihoods).	Qualitative; Narrative research ****
Molla, Mollah… Tomomi (2014) [79]	Quantifying disease burden among climate refugees using multidisciplinary approach: A case of Dhaka, Bangladesh	Dhaka, Bangladesh	(Mobility) Migration. (Climate) Flood/river erosion, drought. (Health) DALYs los, diarrhea, asthma, morbidity.	Quant. desc. *****
Molla, Mollah...Ramasoota (2014) [80]	Multidisciplinary household environmental factors: Influence on DALYs lost in climate refugees community	Dhaka, Bangladesh	(Mobility) Migration. (Climate) Flood/river erosion, drought. (Health) DALYs lost, diarrhea, asthma.	Quant. desc. ***
Murali and Afifi (2014) [81]	Rainfall variability, food security and human mobility in the Janjgir-Champa district of Chhattisgarh state, India	Chhattisgarh state, India	(Mobility) Seasonal & permanent migration. (Climate) Rainfall variability. (Health) Food insecurity, living quality in the city.	Mixed Methods; Case Study & Survey ****
Nawrotzki, Schlak and Kugler (2016) [82]	Climate, migration, and the local food security context: introducing Terra Populus	Burkina Faso & Senegal	(Mobility) International migration. (Climate) Heat waves, droughts, floods. (Health) Food security, child stunting & wasting.	Quant. desc. *****
Oyekale, Oladele and Mukela (2013) [83]	Impacts of flooding on coastal fishing folks and risk adaptation behaviours in Epe, Lagos State	Nigeria	(Mobility) Migration. (Climate) Flooding. (Health) Malaria, typhoid, cholera, diarrhea, dysentery, influenza, tuberculosis.	Quant. desc. **
Pardi, Jungari…Bomble (2020) [84]	Migrant motherhood: Maternal and child health care utilization of forced migrants in Mumbai, Maharashtra, India	India	(Mobility) Forced Migration. (Climate) Drought. (Health) Maternal Child Health care utilisation, immunisation.	Qualitative *
Penning-Rowsell… Thompson (2013) [85]	The ‘last resort’? Population movement in response to climate-related hazards in Bangladesh	Bangladesh	(Mobility) Temporary evacuation. (Climate) Hazard events & disasters. (Health) Ill-health, lack of space & hygiene.	Qualitative ***
Perez-Saez, King… Pascual (2017) [86]	Climate-driven endemic cholera is modulated by human mobility in a megacity	Dhaka, Bangladesh	(Mobility) Urban migration. (Climate) El Nino Southern Oscillation (ENSO). (Health) Cholera.	Modelling
Philibert, Tourigny… Fournier (2013) [87]	Birth seasonality as a response to a changing rural environment (Kayes Region, Mali)	Mali	(Mobility) Seasonal migration. (Climate) Climate and rainfall. (Health) Births registered in primary health care facilities.	Quant. desc. ****
Rademacher-Schulz… Mahama (2014) [88]	Time matters: shifting seasonal migration in Northern Ghana in response to rainfall variability and food insecurity	Northern Ghana	(Mobility) Seasonal & labour migration. (Climate) Rainfall variability. (Health) Livelihood, food security.	Mixed Methods *
Rahaman, Rahman... Hassan (2018) [89]	Health Disorder of Climate Migrants in Khulna City: An Urban Slum Perspective	Khulna City, Bangladesh	(Mobility) Slum migration. (Climate) Climatic disasters (flooding, cyclone, storm surges, sea level rise, river erosion). (Health) Waterborne diseases, undernutrition, micronutrient deficiencies, diarrhea, malaria.	Qualitative **
Rakib, Sasaki…Fukunaga (2019) [90]	Severe salinity contamination in drinking water and associated human health hazards increase migration risk in the southwestern coastal part of Bangladesh	Bangladesh	(Mobility) Migration. (Climate) Sea level rise - Groundwater salinization. (Health) Hypertension, Cardiovascular disease, Renal disease, Diarrhoea, Respiratory disease, Skin disease, Access to healthcare and cost of healthcare.	Mixed Methods ***
Roncoli, Ingram and Kirshen (2001) [91]	The costs and risks of coping with drought: livelihood impacts and farmers’ responses in Burkina Faso	Burkina Faso	(Mobility) Migration. (Climate) Scarce and irregular rainfall, infertile and degraded soils, drought. (Health) Livelihood, food security.	Mixed Methods **
Shanthi, Mahalakshimi and Chandrasekaran (2017) [92]	Assessment of challenges faced by the coastal women due to the impact of climatic change in selected coastal districts of Tamil Nadu, India	Tamil Nadu, India	(Mobility) Urban migration. (Climate) Unusual rainfall, floods, cyclones, change in water quality. (Health) Livelihood, health.	Quant. desc. ***
Suckall, Fraser and Forster (2017) [93]	Reduced migration under climate change: evidence from Malawi using an aspirations and capabilities framework	Malawi	(Mobility) Internal migration. (Climate) Climate stresses (droughts) and shocks (sudden flooding). (Health) Food shortage.	Mixed Methods *
Taiban, Lin and Ko (2020) [94]	Disaster, relocation, & resilience: recovery and adaptation of Karamemedesane in Lily Tribal Community after Typhoon Morakot, Taiwan	Taiwan	(Mobility) Forced Relocation. (Climate) Typhoon. (Health) Food security (food sovereignty), cultural preservation (social capital, mental health).	Qualitative. In-dept interviews and participant observation. ***
Tschakert, Tutu and Alcaro (2013) [95]	Embodied experiences of environmental and climatic changes in landscapes of everyday life in Ghana	Ghana	(Mobility) Rural - urban migration. (Climate) Environmental and climatic change (unpredictable and shifting rainfall). (Health) Well-being, distress.	Qualitative; Phenomenology *****
Van der Geest, Nguyen and Nguyen (2014) [96]	Internal migration in the upper mekong delta, viet nam: what is the role of climate related stressors?	Vietnam	(Mobility) Internal migration. (Climate) Climate-related stressors (floods, storms, rainfall). (Health) Food insecurity.	Mixed Methods ***
Van der Geest, Burkett... Wheeler (2020) [97]	Climate change, ecosystem services and migration in the Marshall Islands: are they related?	Marshall Islands	(Mobility) International migration. (Climate) Sea-level rise, drought, extreme heat. (Health) Access to healthcare, Water security.	Mixed Methods ****
Warner and Afifi (2014) [98]	Where the rain falls: Evidence from 8 countries on how vulnerable households use migration to manage the risk of rainfall variability and food insecurity	Guatemala, Peru, Ghana, Tanzania, Bangladesh, India, Thailand & Vietnam	(Mobility) Migration. (Climate) Rainfall variability. (Health) Food insecurity (food production & market food availability).	Mixed Methods ****
Wolsko and Marino (2016) [99]	Disasters, migrations, and the unintended consequences of urbanization: What’s the harm in getting out of harm’s way?	Shishmaref, Alaska	(Mobility) Planned relocation. (Climate) Erosion, wind, ice melt, floods. (Health) Mental health status.	Qualitative; Ethnography **
Woodhall-Melnik and Grogan (2019) [100]	Perceptions of mental health & wellbeing following residential displacement & damage from the 2018 St. John River Flood	Canada	(Mobility) Residential displacement. (Climate) Flood. (Health) Mental health, well-being.	Qualitative *****
Zaman, Sammonds… Rahman (2020) [101]	Disaster risk reduction in conflict contexts: Lessons learned from the lived experiences of Rohingya refugees in Cox’s Bazar, Bangladesh	Bangladesh	(Mobility) Forced displacement. Trapped populations. (Climate) Landslides, tropical cyclones, flash-flooding. (Health) Infectious disease outbreaks. Access to healthcare. Food and water security.	Mixed Methods ***

Individual Quality Appraisal. The ranking system is according to the articles adherence to each of the criteria i.e—frequency (%) we could answer ‘yes’ to the quality appraisal question in MMAT and not ‘no’ or ‘cannot tell’ (***** 100%, **** 80%, *** 60%, ** 40% * ≤20%). In mixed methods studies, there were 15 quality appraisal (QA) questions (quantitative component, qualitative component, mixed methods general) and we used the overall score (denominator 15 not 5). QA Collated by method: When we refer to QA overall being ‘high’ in the text, we refer to the frequency (%) with which we could answer ‘yes’ to the QA questions for that method group. Very high (81%–100%) High (61%–80%) Good (41%–60%) Fair (21%–40%) Poor (0%–20%).

**Table 4 ijerph-17-09342-t004:** Synopsis of themes and illustrative quotes.

Theme	Illustrative Quote
1. Avoid the universal promotion of migration as an adaptive response to climate risks. Prevent forced migration by investing in climate change adaptation, disaster risk reduction and sustainable development. Consider planned relocation as a last resort.	*‘The policy implication is that governments should not make assumptions a priori about whether a location is undesirable and promote migration as a blanket solution to the negative impacts of climate change’—Quantitative descriptive, Peru* [12].
2. Preserve cultural and social ties of mobile populations. Strengthen governance of socio-ecological systems.	*‘Developing government frameworks that can draw on the strengths of the community-led approaches to relocation whilst also providing a mechanism for communities to stay intact will be an important step forwards for Small Island Developing States (SIDs) facing these climate pressures’—Qualitative case studies, Alaska & Solomon Islands* [40].
3. Enable participation of migrants in their sites of relocation and resettlement. Support the self-sufficiency of both incoming and host communities by supporting new livelihoods, developing social networks, and integrating cultural considerations.	*‘As to the migrants, their value added to the areas of destination should be maximized by involving them in activities that match their skills (e.g., farming and construction), so that they actively contribute to the overall welfare of the new areas on one hand and to the well-being of themselves and their families on the other’—Mixed methods, Tanzania* [38].
4. Strengthen health systems and reduce barriers for migrants to access health care.	*‘Relocation of local health facilities with basic and emergency care provisions to areas in which the displaced have resettled, reinforcement of Family Planning services, and extension of coverage of the Maternity Allowance benefits in the displacement prone mainland riverine areas are recommended policy responses’—Cross sectional survey, Bangladesh* [59].
5. Provide migrants with the requirements and the determinants for good health.	*‘The rights of these displaced people, including the right to health, are often poorly protected in practice. More vigorous application of existing human instruments is needed, as well as clarification and possibly re-definition of the rights of those displaced’—Qualitative study, Bangladesh* [89].
6. Integrate health into loss and damage assessments. Consider people at higher risk, including those that are immobile or trapped.	*‘The findings outlined a long line of climate-induced non-economic losses and damages that people faced through the rural-urban move from the island, and through the displacement in the slum. These included the loss of identity, honour, sense of belonging, physical and mental health or wellbeing’—Mixed Methods, Bangladesh* [32].

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
