# Peer review of "A Meta-Synthesis of Policy Recommendations Regarding Human Mobility in the Context of Climate Change"

_ijerph, 2020, doi:10.3390/ijerph17249342_

Round 1
Reviewer 1 Report
This is an interesting paper which seeks to undertake Systematic searches in four academic databases (PubMed, Ovid Medline, Global Health and Scopus) and Google Scholar for empirical studies published between 1990 – 2020 that used any study design to investigate migration and health in the context of climate change. Despite the noted strengthens, just a couple of minor observations that would indeed enhance the quality of the revised manuscript.
Justification for the selection of the data bases- The authors should include detailed justification for the selection of the databases underpinned by literature. For instance, what are the benefits of using several search engines to find articles which led the authors to that? Some studies suggest such an approach would minimise the bias and helps to refine a broad range of articles.
Literature review – There are some strengths and depth within the studies reviewed. However, some key studies such as Zuo et al. (2015) established that Heat waves have significant impacts on both ecosystems and human beings are missing from this review. Please revisit this publication as it is a good source as the study reported on the effects of heatwaves or temperatures on mortality and other aspects of change-related health impacts.
Zuo, J., Pullen, S., Palmer, J., Bennetts, H., Chileshe, N. and Ma, T. (2015), “Impacts of heat waves and corresponding measures: a review” Journal of Cleaner Production, Vol. 92, pp. 1-12.
Methods – The SLR analysis such as descriptive analysis of the literature and thematic or content analysis of the themes should have been specified within the methods section.
Author Response
Please see response to reviewer's table attached.

Reviewer 2 Report
First of all, I would like to congratulate the authors for this interesting work. It is very well presented and few modifications should be made. Just as a suggestion, maybe table 3 can be presented as an appendix instead of inside the text and figure 5 could be presented with less colour to make it easier to read.
Author Response
Thank you for this constructive feedback.
Please find the response to reviewers table attached.
Kind Regards,
Trish.
